

# Global Downscaling of Remotely-Sensed Soil Moisture using Neural Networks

Seyed Hamed Alemohammad[1,2], Jana Kolassa[3,4], Catherine Prigent[1,2,5], Filipe Aires[1,2,5], Pierre Gentine[1,2,6]

[1]Department of Earth and Environmental Engineering, Columbia University
[2]Columbia Water Center, Columbia University
[3]Universities Space Research Association, Columbia, MD
[4]Global Modelling and Assimilation Office, NASA Goddard Space Flight Center, Greenbelt, MD
[5]Observatoire de Paris
[6]Earth Institute, Columbia University

*Correspondence to*: Pierre Gentine (pg2328@columbia.edu)

**Abstract.** Characterizing soil moisture at spatio-temporal scales relevant to land surface processes (i.e. of the order of a kilometer) is necessary in order to quantify its role in regional feedbacks between land surface and the atmospheric boundary layer. Moreover, several applications such as agricultural management can benefit from soil moisture information at fine spatial scales. Soil moisture estimates from current satellite missions have a reasonably good temporal revisit over the globe (2-3 days repeat time); however,
their finest spatial resolution is 9km. NASA's Soil Moisture Active Passive (SMAP) satellite estimates soil moisture at two different spatial scales of 36km and 9km since April 2015. In this study, we develop a neural networks-based downscaling algorithm using SMAP observations and disaggregate soil moisture to 2.25km spatial resolution. Our approach uses mean monthly Normalized Differenced Vegetation Index (NDVI) as an ancillary data to quantify sub-pixel heterogeneity of soil moisture. Evaluation of the downscaled soil moisture estimates against in situ observations shows that their accuracy is better than or equal
to the SMAP 9km soil moisture estimates.

## 1 Introduction

Soil moisture is a key variable constraining the fluxes between the land surface and atmosphere boundary, and therefore it plays a key role in regulating the feedbacks between the terrestrial water, carbon and energy cycles (Berg et al., 2014; McColl et al., 2017; Seneviratne et al., 2010). Soil moisture partitions the surface energy between latent heat and sensible heat fluxes (Entekhabi et al.,
1996; Gentine et al., 2007, 2011; Koster et al., 2010). Moreover, plant photosynthesis is regulated by the water available to them through their roots along with atmospheric conditions (Rodríguez-Iturbe and Porporato, 2007; Seneviratne et al., 2010; Volk et al., 2000). Finally, soil moisture regulates the surface water fluxes including infiltration and surface runoff generation (Salvucci, 1993; Salvucci and Entekhabi, 1994; Sun et al., 2011). Therefore, there is a need to characterize soil moisture at spatial scales relevant to the representation of land-surface and mesoscale processes in the atmosphere. This can potentially improve representation of
evapotranspiration, runoff and precipitation in hydrologic and weather prediction models and result in improved predictive skills (Gedney and Cox, 2003). In addition, knowledge of soil moisture at fine spatial scales is necessary to improve farming practices and optimizing irrigation scheduling.

Soil moisture spatial variability is regulated by several factors including but not limited to precipitation, soil texture, surface vegetation and topography. A combination of these factors results in high spatial heterogeneity for soil moisture (Blöschl and
Sivapalan, 1995; Famiglietti et al., 2008; Manfreda et al., 2007; Peng et al., 2017; Western and Blöschl, 1999).

Current global soil moisture estimates are available at coarse spatial scales (between 9 and 40 km) which limits their suitability for applications such as evapotranspiration modeling, particularly in regions with high atmospheric convection as well as farm



management. Therefore, several approaches have been introduced to downscale soil moisture retrievals at finer resolution (~1km) (Hatfield, 2001; Mascaro et al., 2011; Merlin et al., 2006, 2008c; Peng et al., 2017; Piles et al., 2011; Srivastava et al., 2013). Here, our objective is to generate a global downscaling product using Soil Moisture Active Passive (SMAP) measurements (Chan et al., 2016; Colliander et al., 2017b; Entekhabi et al., 2010).

Soil moisture downscaling can be conducted through different strategies. The first one is to use the synergy of active and passive observations to take advantage of highly accurate but coarse resolution passive observations and active measurements available at higher spatial resolution but not as directly related to soil moisture as passive microwave measurements (Das et al., 2011; Jagdhuber et al., 2015, 2016; Leroux et al., 2016; Montzka et al., 2016; Njoku et al., 2002; Piles et al., 2009; Wu et al., 2017). However, due to the failure of SMAP active instrument in July 2015 these approaches are not directly applicable to SMAP.

Another approach to downscaling is to use ancillary data in combination with coarse scale soil moisture estimates to describe the spatial heterogeneity of soil moisture. Several studies have used this approach to develop soil moisture at fine spatial scales (Chakrabarti et al., 2016, 2017; Mascaro et al., 2011; Piles et al., 2011, 2014; Srivastava et al., 2013; Verhoest et al., 2015). However, there are limitations in these techniques. Some of them use linear relationships (i.e. a projection) to define the impact of spatial heterogeneity using ancillary data, typically in combination with a radiative-transfer model to relate surface temperature

and soil moisture (Colliander et al., 2017a; Merlin et al., 2005, 2008a, 2008b, 2008c). A major issue is that surface temperature at finer spatial scales from satellites cannot be estimated under cloudy conditions. The other group of downscaling methods use complex and computationally intensive disaggregation algorithms that are typically unsuitable for global applications. Another group of methods use physical land surface models at fine spatial scales as an information useful for downscaling which adds significant uncertainty to the downscaling due to the errors of the model parameter estimates (Ines et al., 2013; Merlin et al., 2006;

Roerink et al., 2000; Shin and Mohanty, 2013). Peng et al. (2017) provide a more detailed review of the current status of soil moisture downscaling algorithms and their advantages and disadvantages.

Our approach in this study is to use Neural Networks (NN) to develop a downscaling algorithm for SMAP soil moisture estimates at the global scale. The capability of NNs in learning complex relationships between inputs and target data as well as their quick run-time after training are among the reasons for their popularity in Earth science and remote sensing problems. In recent years,

NN has been used to develop soil moisture retrieval algorithms from either passive or active instruments or combination of both (Aires et al., 2012; Alemohammad et al., 2017a; Jiménez et al., 2013; Kolassa et al., 2013, 2016, 2017a, 2017b; Rodríguez-Fernández et al., 2017; Rodríguez-Fernández et al., 2015). They have also been used to retrieve surface turbulent fluxes from remote sensing observations (Alemohammad et al., 2017b; Jiménez et al., 2009).

Our final product is soil moisture estimates at 2.25km spatial resolution with full global coverage every 2-3 days as dictated by the

SMAP orbit from April 2015 until end of March 2017 (the first two years of SMAP mission). Moreover, we use ancillary data that are available at global scale at all times (monthly NDVI and topographic index) to mitigate issues with current downscaling algorithms.

## 2 Datasets

In order to perform the downscaling algorithm, we use three remote sensing based observations: (1) SMAP estimated coarse-

resolution soil moisture (Section 2.1), (2) Normalized Difference Vegetation Index (NDVI) as a vegetation index that is strongly correlated with soil moisture spatial patterns (Section 2.2) and (3) a topographic index to account for the location of water in the landscape (Section 2.3).





These data are provided on different spatial grids and are re-projected to a common grid for our analysis (Section 3.1). Moreover, in situ soil moisture data that are used for evaluation are introduced in Section 2.4.

**2.1 SMAP Soil Moisture**

SMAP satellite estimates surface soil moisture (within the top 5cm of the soil) across the globe with a 2-3 days repeat time
(Entekhabi et al., 2010). SMAP uses a passive microwave radiometer in L-band (1.4 GHz) which is known to be sensitive to surface soil moisture and transparent to clouds and atmospheric moisture. Brightness temperature observations from SMAP are used together with ancillary data on vegetation condition and surface temperature to estimate soil dielectric constant from a zeroth-order radiative transfer model, commonly referred to as τ-ω model (Jackson, 1993; Jackson and Schmugge, 1991; Kurum et al., 2011). Finally, soil dielectric constant estimates are converted to volumetric soil moisture estimates using soil texture data (Chan
et al., 2016; O'Neill et al., 2015). Measurements from SMAP are available from April 2015 to present.

In this study, we use two SMAP Level-3 (L3) passive soil moisture products. The first is the version 4.0 SPL3SMP product, which estimates soil moisture on a 36km the Equal-Area Scalable Earth Grid (EASE-grid 2.0) (O'Neill et al., 2016b). This product provides soil moisture estimates for both the morning (6 AM) and evening (6 PM) overpasses of the satellite; however, we only use the morning overpasses in this study since it minimize observation errors due to Faraday rotation and soil and canopy
temperatures are in equilibrium in the morning. The second is the version 1.0 SPL3SMP_E product, which estimates soil moisture with an enhanced spatial resolution posted on 9km EASE-grid 2.0 (O'Neill et al., 2016a). Both of the products use the same brightness temperature observations from SMAP radiometer; however, in the enhanced product spatial resolution is enhanced using the Backus Gilbert interpolation technique (Chan et al., 2017; Chaubell et al., 2016). While the SPL3SMP_E product is posted on a 9 km EASE 2.0 grid, its native resolution is coarser (~ 33 km). Chan et al., 2017, provides a detailed explanation of
how the native resolution and grid spacing of this product are different. Both products are downloaded from National Snow and Ice Data Center (NSIDC) at https://nsidc.org/data/smap/smap-data.html .

To be consistent with the SMAP grid, we re-project our ancillary data to EASE-grid 2.0 at the respective spatial resolutions. Moreover, our final downscaled soil moisture estimates are at 2.25km spatial resolution on the EASE-grid 2.0 which is nested within the 9km and 36km grids of SMAP L3 soil moisture estimates (Section 3.1).

**2.2 MODIS NDVI**

Plant's photosynthesis and subsequently vegetation greenness are regulated by the moisture available to plants through their roots, as well as atmospheric conditions (Béziat et al., 2013; Kool et al., 2014; Wang et al., 2014). In turn, vegetation modifies its soil moisture environment through changes in the partitioning of evapotranspiration and through precipitation interception (Coenders-Gerrits et al., 2013; Markewitz et al., 2010). Therefore, surface vegetation cover can be a proxy for the spatial heterogeneity of soil
moisture. Vegetation greenness indices such as NDVI are estimated by observations at red and near-infrared frequencies and have relatively high spatial resolution. In this study, we use monthly mean values of NDVI from the MODerate resolution Imaging Spectroradiometer (MODIS) instrument on board Terra satellite. We use version 006 of MYD13A3 product that provides vegetation indices on a 1km spatial resolution at global scale (Didan, 2015).

MODIS estimates are provided on a sinusoidal grid, and due to its high resolution, data are divided into 10° × 10° tiles. In order to
re-project these data to SMAP EASE-grid 2.0, we used two open source libraries in Python. First, the pyModis library was used to mosaic all the tiles together and generate one global map for each measurement. Next, GDAL library was used to reproject MODIS estimates from its original 1km sinusoidal grid to the EASE-grid 2.0s.

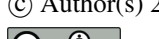



### 2.3 Topographic Index

Topographic index (TI) or topographic wetness index is a measure of the soil's saturation tendency given the upstream drainage area and the slope of the local outflow (Marthews et al., 2015). This index is a good proxy for the heterogeneity of soil moisture at the watershed scale and thus could be expected to provide useful information on landscape moisture organization. Here, we use

TI at 15 arcsec spatial resolution developed by Marthews et al., 2015. We upscaled the TI values to the EASE-grid 2.0 at 36km, 9km and 2.25km to be used as an ancillary data.

### 2.4 International Soil Moisture Network (ISMN)

In situ soil moisture observations from a set of local networks are provided through the ISMN network and are used to independently evaluate the performance of our downscaling algorithm. ISMN collects and standardizes soil moisture observations

from several networks around the globe (Dorigo et al., 2011, 2013). The spatial density of probes and their measurement depth varies across different networks. In this study, we only use data from networks that have measurement at a depth of 5cm below surface (the nominal penetration depth of SMAP) during the first two years of the SMAP mission. Moreover, we only keep pixels that have at least 20 measurements collocated in time with SMAP observations which are 6am local time. As a result, we only use data from the following 10 networks: FMI (8 pixels), iRON (4 pixels), REMEDHUS (20 pixels), RSMN (19 pixels), SCAN (159

pixel), SMOSMANIA (17 pixels), SNOTEL (344 pixels), SOILSCAPE (10 pixels), TERENO (5 pixels), and USCRN (98 pixels) (Albergel et al., 2008; Bell et al., 2013; Calvet et al., 2007; Moghaddam et al., 2010, 2016; Zacharias et al., 2011).

### 3 Methodology

Our downscaling algorithm is based on an NN approach that uses the two soil moisture estimates from SMAP at 36km and 9km for training, and then retrieves soil moisture at 2.25km spatial resolution using the 9km estimates from SMAP. The NN relates the

coarse scale soil moisture and NDVI estimates as well as the fine scale NDVI estimates as input to the fine scale soil moisture estimates as output. To do so, we assume that the scaling relationship between 36km and 9km soil moisture estimates is the same as the scaling relationship between 9km and 2.25km estimates. To the best of our knowledge, this is the first time that the assumption of similar scaling relationship is used to downscale soil moisture, while it has been shown that soil moisture has a fractal scaling property across scales (Famiglietti et al., 2008). In the following the spatial grid setup, data preprocessing and NN

training are explained.

### 3.1 Spatial Grids

SMAP observations are provided on the EASE-grid 2.0 at different spatial resolutions. The advantage of this grid is the possibility of having nested grids at different spatial resolutions. In this case, SMAP's original radiometer-only soil moisture estimates are at 36km resolution and the enhanced estimates are at 9km resolution which is nested within the 36km one. Since we use the

relationship between the 36 and 9km grids (¼ of the original scale) to train our NN algorithm, our final downscaled estimates also have a spatial resolution of ¼ of the 9km product which is used as input in the retrieval step. This results in a spatial resolution of 2.25km on the EASE-grid 2.0. Figure 1 shows the setup of grids used in the training and retrieval steps.




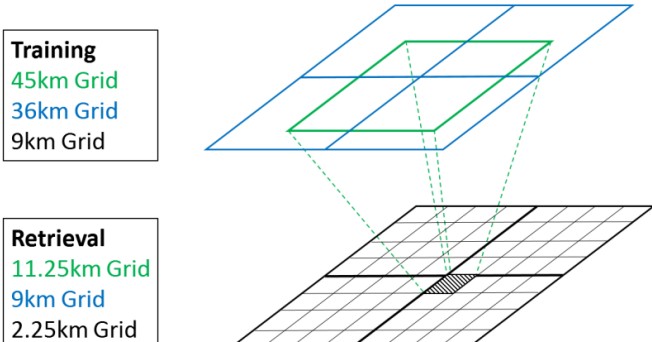

**Figure 1-** Two levels of spatial grids used for training and retrieval steps in the NN algorithm. While both steps have similar grid structures, the spatial resolutions are different as listed here.

To the best of our knowledge, EASE-grid 2.0 at 2.25km has not been used in any other product before. Therefore, we defined the grid using the same parameters as for 36km and 9km, listed in (Brodzik et al., 2012, 2014), other than the grid spacing which is set to 2252.0138025365 m (equals to ¼ of the grid spacing for the 9km grid).

Our NN algorithm estimates soil moisture at a given pixel of the finer resolution grid (the hatched pixel on the black grid in Figure 1). To generate the coarse scale soil moisture estimates that are used as input to the NN algorithm, we use a moving window

averaging at the coarser resolution grid (the green pixel in Figure 1) that is centered on the target pixel at the finer grid. In the training step, this moving window is 45 km (applied over the 36 km product) and in the retrieval step, the moving window is 11.25km (applied over the 9 km product). All the inputs to the algorithm are averaged at the scale of the moving window from the coarse resolution grid using an area-weighted averaging. The only disadvantage of the moving window technique is that we cannot retrieve soil moisture at the finer scale near coastal regions or places where SMAP does not provide estimates of soil moisture

(such as urban areas, ice covered regions and frozen soil conditions). While this is a limitation of our approach, its impact on the spatial coverage of our downscaled soil moisture estimates is very limited geographically. This limitation can potentially be improved by incorporating an extrapolation algorithm that uses NDVI to estimate soil moisture in coastal pixels (Aires et al., 2017). Moreover, we used the static water bodies' mask from SMAP 9km soil moisture product and masked out all pixels that have a water fraction of more than 10%. The reason is that high water fraction impacts brightness temperature observations and

results in a biased estimate of soil moisture. We also observed higher discrepancies between the SMAP 36km and 9km soil moisture estimates in regions with high water fraction (not shown here).

### 3.2 Neural Networks Setup

Our retrieval algorithm is a statistical approach based on NN. NN uses a set of training data to learn the relationship between inputs and outputs without explicitly modelling the physical relationship between them. This is a powerful statistical approach and has

been shown to be able to approximate any continuous function (Cybenko, 1989). NN consists of a set of layers and neurons in each layer that mimic the neural system in humans' brain. Each neuron has a weight and bias corresponding to the neurons from its previous layer. The output of a neuron is a linear summation of the weighted inputs plus the bias that goes through an activation function. In each Multi-Layered Perceptron (MLP) NN (Rumelhart et al., 1985), there is an input layer, an output layer and one or more hidden layers in between. Number of neurons in the input layer is equal to the number of inputs provided, and number of

neurons in the output layer is equal to the number of outputs given to the NN during training.





The right choice of the number of neurons in the hidden layer and number of hidden layers depend on the complexity of the relationship between inputs and outputs. While having more hidden layers and neurons can potentially increase estimation accuracy, it might result in over-fitting to training data. This means that the NN has lower accuracy when applied to other independent data. To avoid over-fitting, we use the simplest network (i.e., minimum of hidden layers and neurons) that has an

acceptable estimation accuracy. Moreover, we use an extensive and representative set of data for training that covers the full range of soil moisture dynamics across different climates and surface conditions. Furthermore, we test the generalization ability of the NN in an independent dataset.

We use the first two years of data from SMAP beginning on April 1st, 2015 until March 31st, 2017. For training data, we sample SMAP soil moisture estimates every 10 days at global scale. This results in 46.3 million data points after removing pixels with

high percentage of water bodies. For validation, we sample the SMAP data every 5 days with one day lag with respect to the training data at global scale; therefore, training and validation samples are mutually exclusive. This results in 92 million data points after removing pixels with high percentage of water bodies.

During training the weights and biases of each neuron are estimated iteratively by minimizing a mean squared error cost function using a gradient-descent algorithm with back propagation (Hagan and Menhaj, 1994; Rumelhart et al., 1985). For this purpose,

training data are divided to three categories of training (60%), validation (20%) and testing (20%). After each iteration on the training category, the validation data are used to check for over-fitting and test data are used to check convergence. When changes in the cost function are smaller than a threshold, training stops, and weights and biases that resulted in the best performance are selected as the parameters of the network. The validation data used here should not be mistaken with the independent validation dataset sampled mutually exclusive to the training dataset. Throughout the paper, we will use "validation data" terminology to

refer to the dataset sampled mutually exclusive to the training data (explained in the previous paragraph) and will be used to evaluate the performance of the NN training in terms of over-fitting.

We trained a set of networks with number of hidden layers between 1 and 5 and number of neurons in each hidden layer between 1 and 15. After evaluating their performance, we selected the network with one hidden layer and 5 neurons, which has a good accuracy, while adding more hidden layers or neurons did not change the performance. We use the hyperbolic tangent sigmoid

function as the activation function for the hidden layer, a standard practice, and a linear function as the activation function for output layer.

### 3.3 Inputs and Downscaling Schemes

We develop four different schemes with increasing number of ancillary inputs for downscaling SMAP soil moisture estimates. In this section, we introduce each of them, and we compare their performance in Section 4. At the end, only one scheme was selected

to provide the final downscaled soil moisture estimates.

Scheme R1 has the three following inputs: soil moisture estimates from SMAP on the moving window grid at the coarse scale resolution (45km for training and 11.25km for retrieval), mean NDVI at the coarse scale moving window, and NDVI in the target pixel at the fine scale grid (9km for training and 2.25km for retrieval).

Scheme R2 has all the inputs in scheme R1 plus standard deviation of NDVI at the fine scale pixels within the moving window.

For example, in the training step there are 25 9km pixels within the 45km moving window. We estimate the standard deviation of those, and input that to the network. This estimate provides a proxy of the heterogeneity within the coarse scale grid.

Scheme R3 has all the inputs in scheme R1 plus TI at the moving window of the coarse scale and TI in the target pixel at the fine scale grid.



Scheme R4 has all the inputs from schemes R1-R3. Table 1 lists all the four schemes and their respective inputs. For each scheme, the 9km SMAP soil moisture estimates are used as the NN target data.

**Table 1- Inputs used in each of the downscaling schemes. The output is always soil moisture estimates at 2.25km resolution.**

| # | Input 1 | Input 2 | Input 3 | Input 4 | Input 5 | Input 6 |
|---|---|---|---|---|---|---|
| R1 | SM at 45 km | NDVI at 45 km | NDVI at 9 km for target pixel | -- | -- | -- |
| R2 | SM at 45 km | NDVI at 45 km | NDVI at 9 km for target pixel | $\sigma_{NDVI}$ at 45 km for all pixels at 9 km | -- | -- |
| R3 | SM at 45 km | NDVI at 45 km | NDVI at 9 km for target pixel | TI at 45 km | TI at 9km for target pixel | -- |
| R4 | SM at 45 km | NDVI at 45 km | NDVI at 9 km for target pixel | TI at 45 km | TI at 9km for target pixel | $\sigma_{NDVI}$ at 45 km for all pixels at 9 km |

## 4 Results

In this section, we first present the results of NN training and evaluation of the downscaling from 36km to 9km. We only present the results from Scheme R1, since all four schemes have similar performances. Next, we apply the downscaling scheme to the entire 2-year SMAP data record and generate soil moisture at 2.25km spatial resolution. Finally, we evaluate the accuracy of downscaled soil moisture from all the four schemes using in situ soil moisture estimates from the ISMN dataset.

### 4.1 Evaluating NN Training

To evaluate the success of the NN training, we compute the correlation coefficient ($R^2$) and the unbiased Root Mean Square Difference (ubRMSD) between the NN estimates and the target data for the training and validation data. We compare the metrics aggregated across all the data and spatially at each pixel.

Figure 2 shows the density scatterplot of 9km NN estimates versus target 9km soil moisture data in the training and validation datasets. Both datasets have similar ubRMSD and $R^2$ which shows that the NN is able to generalize beyond the training data. $R^2$

of ~0.98 is an almost perfect correlation between the target and estimates, and indicates that the NN setup with the inputs provided is capable of learning the relationship between coarse and fine scale soil moisture.

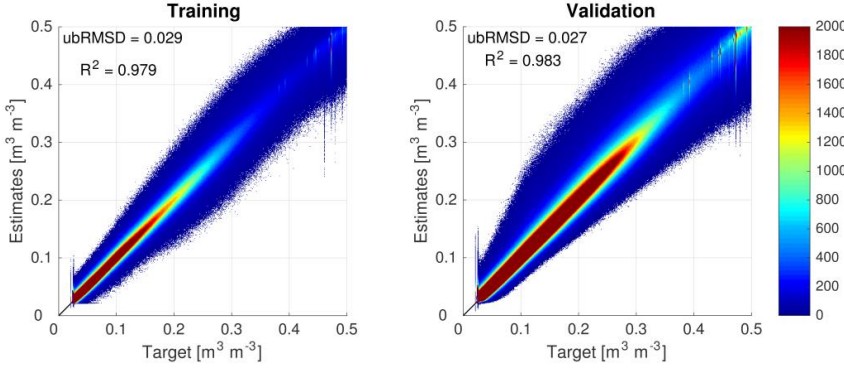

**Figure 2- Density scatterplot of NN estimates vs target data for training (left) and validation (right) datasets. Color shades show number of samples in each part of the scatterplot.**




In order to compare the performance of NN with other techniques, we present similar results for two other approaches. The first approach assumes a uniform distribution of soil moisture (i.e. no heterogeneity) within each 36km grid pixel. This means that all the 9km pixels that fall within a 36km grid pixel are attributed the same soil moisture value as the 36km one (i.e. no downscaling). The second approach is to use a linear interpolation (i.e. a crude downscaling). This is used as a reference to a simpler approach than the NN. In this approach, soil moisture values on the 36km grid are assumed to be at the center point of the pixel. Then using a linear interpolation, soil moisture is estimated at the center point of each 9km grid pixel. We implement the interpolation at the global scale for each SMAP observation. Figure 3 shows the density scatterplots for these two approaches. As the scatterplots and performance metrics show, the NN algorithm has higher correlation and lower ubRMSD compared to both no heterogeneity and interpolation approaches. As expected, the homogeneous algorithm has the worst performance.

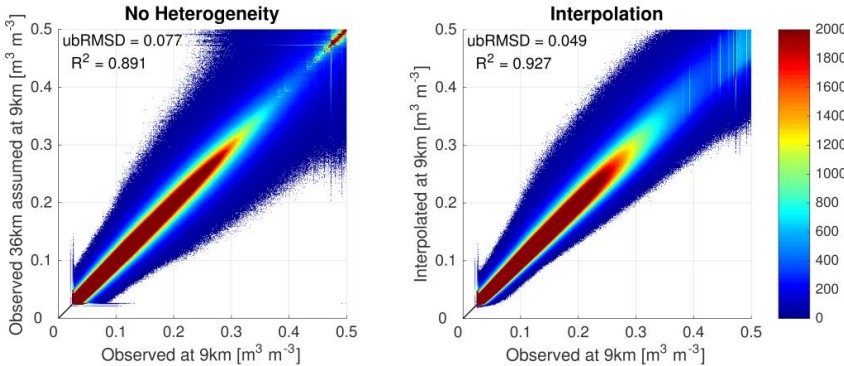

**Figure 3- Density scatterplots of uniform 9km soil moisture estimate (left) and interpolated soil moisture at 9km (right) vs observed 9km soil moisture. Color shades show number of samples in each part of the scatterplot.**

To further evaluate the performance of the NN downscaling algorithm, we evaluate the correlation between target soil moisture and NN estimates at 9km in each pixel (Figure 4). These estimates are based on the validation dataset, and on average 50 data points are used to calculate the correlations (in some snowy regions like Himalaya Mountains as few as 20 data points were used, the minimum number of samples set to estimate the correlation in this evaluation). Correlations range between a minimum of 0.6 in dense tropical forests (where NDVI saturates and limited spatial inhomogeneity can be found) to ~1 in most other parts of the world, except in very northern regions were the correlation is closer to 0.9. The smaller value of correlation in the tropics is a result of the low seasonality and saturation effect of NDVI in those regions, so that NDVI does not provide useful information for downscaling (Morton et al., 2014). These regions have relatively high soil moisture and relatively low variability in NDVI throughout the year. In addition, NDVI and EVI tend to saturate at high vegetation cover so that there is only limited variability in those regions. As a result, disaggregating soil moisture using NDVI has a lower accuracy compared to other regions. Nevertheless, NDVI and EVI are the only global fine scale vegetation indices that can be used as ancillary data to describe the sub-pixel heterogeneity of soil moisture, and both lack seasonality in the dense tropical forests. Moreover, Figure S1 shows the percentage difference between the NN retrieval and SMAP estimates at 9km. Largest differences are observed in mountainous and coastal regions where soil moisture variations are influenced by factors such as the terrains lope. However, inclusion of other ancillary data such as slope in other downscaling schemes did not improve the bias and schemes R2-R4 have similar performance both in terms of aggregated $R^2$ and ubRMSD and spatial patterns of correlation between NN estimates and target soil moisture.

Moreover, Figures S2-S5 show the spatial correlation maps and percentage difference between SMAP estimates at 9km and the estimates from Interpolation and No Heterogeneity approaches. These figures show that both approaches have significantly higher bias with respect to the SMAP estimates compared to the NN retrieval.





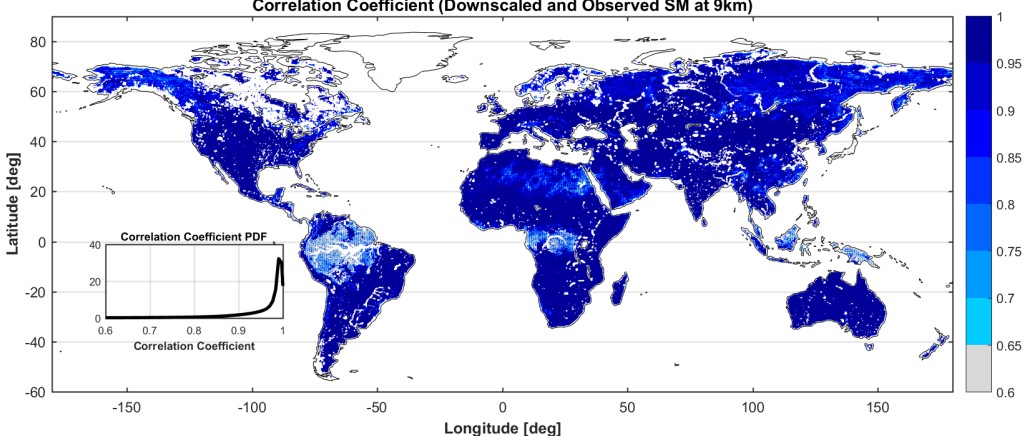

**Figure 4- Correlation coefficient ($R^2$) between SMAP observed soil moisture at 9km and NN downscaled soil moisture at 9km. White regions indicate no data.**

### 4.2 Downscaled Soil Moisture

Using the NN that was trained and evaluated in the previous section, we then estimate soil moisture at 2.25km EASE-grid 2.0 resolution. For this, we use the SMAP 9km observed soil moisture as input together with ancillary data for each scheme as described in Section 3.3.

Figure 5 shows the average global soil moisture at three different spatial resolutions (36km and 9km from SMAP observations, and 2.25km downscaled from SMAP observations). Overall, the spatial patterns are as expected, with some heterogeneity being

added as the resolution increases. These maps show that our downscaling algorithm preserves the large-scale spatial patterns of soil moisture while disaggregating it at local scales. Moreover, the latitudinal average plots (on the right side of each panel of Figure 5) show that at higher spatial resolutions there is more spatial heterogeneity. The latitudinal average for the 36 km product is much smoother than the 2.25 km one.

### 4.3 Large Scale Evaluation of Downscaled Soil Moisture

In this section, we analyze large-scale patterns of the downscaled soil moisture estimates at 2.25km spatial resolution. Our first analysis is the spatial heterogeneity of the downscaled soil moisture. We calculate coefficient of variation (CV), which is the spatial standard deviation divided by the mean, of the 16 2.25km pixels within each 9km pixel at each time. Figure 6 top panel shows the mean CV at each pixel for the downscaled soil moisture. For comparison, we also calculate CV for the 9km soil moisture estimates from SMAP at the 36km grid (Figure 6 bottom panel). The two panels in Figure 6 have different range of CV which is expected

given the difference in their spatial scales. However, they reveal similar spatial patterns. This means that in regions that have high spatial heterogeneity in the 9km SMAP estimates, we also see high heterogeneity in the 2.25km downscaled soil moisture estimates. Therefore, our NN algorithm is appropriately explaining the spatial variability of soil moisture using NDVI as ancillary data. The other downscaling schemes (R2-R4) exhibit similar spatial patterns for CV. This is an indication that the other inputs to these schemes do not provide any extra information on the heterogeneity of soil moisture at the finer spatial scale.






**Figure 5- Average soil moisture at 36km (top), 9km (middle) and 2.25km (bottom) between April 2015 and March 2017. Shaded plot on the right side of each panel shows the mean and one standard deviation of soil moisture at each latitude.**

5    Our second analysis evaluates the conservation of the water balance within each coarse scale pixel. Since we are disaggregating soil moisture from 9km grid to 2.25km grid, the downscaling algorithm may not preserve the mean soil moisture within each coarse scale pixel (water balance). Figure 7 shows the spatial map of the percentage average difference (with respect to their mean) between the mean soil moisture from the 16 2.25km pixels within each 9km pixel and soil moisture estimates at 9km from SMAP. The inset PDF also shows the probability distribution of these differences. On average, there is a positive bias of ~1.5% in the





differences, and the downscaled soil moisture estimates at 2.25km are slightly positively biased, i.e. more humid. However, the differences are small (mostly less than 3% in absolute value) and within the SMAP retrieval mission accuracy (<0.04 m$^3$ m$^{-3}$). This analysis also reveals that our downscaling algorithm is robust in preserving the water balance within each coarse scale pixel and still within the original requirements of the SMAP mission.

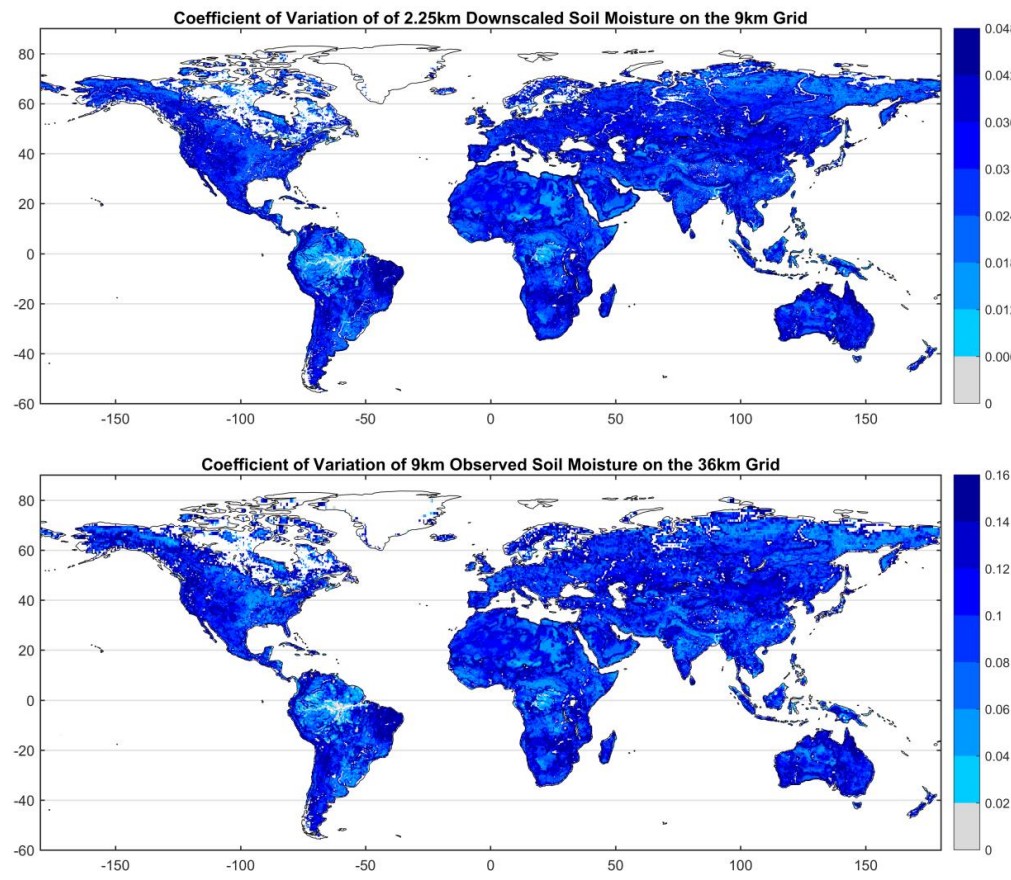

**Figure 6- Coefficient of variations of 2.25km downscaled soil moisture within each 9km grid pixel (top), and of the 9km observed soil moisture within each 36km grid pixel (bottom).**

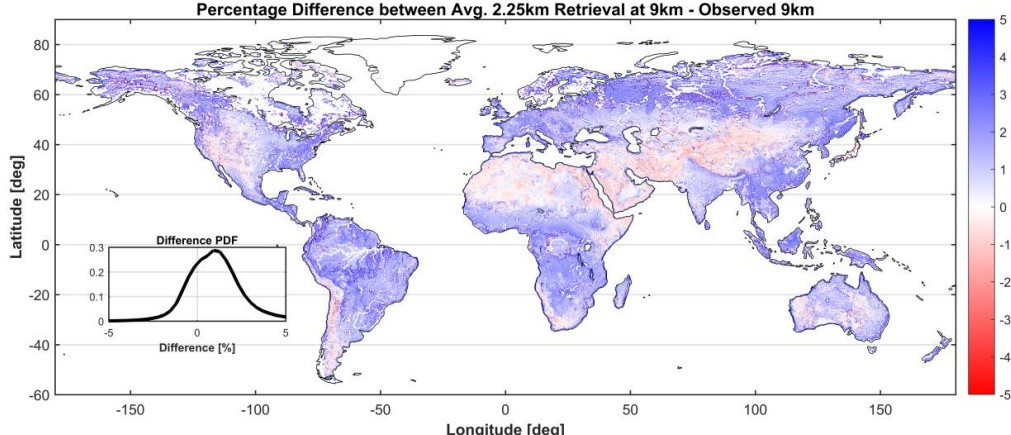

**Figure 7- Percentage difference with respect to mean of average 2.25km soil moisture estimates at 9km and observes 9km soil moisture.**





### 4.4 Comparison against ISMN Data

In this section, we evaluate downscaled soil moisture estimates against ISMN dataset. We calculate $R^2$, anomaly $R^2$ and ubRMSE for each of the four downscaling schemes as well as SMAP estimates at 36km and 9km. Anomalies are calculated based on a 30 day moving average window. We use stations from 10 networks within the ISMN dataset, and Figure S6 shows the spatial distribution of these stations. For networks that have more than one station within each SMAP pixel, we average the stations within the 2.25km pixels and then calculate the metrics for each of the 36km, 9km and 2.25km products. Figure 8 shows the average of each metric across stations for different networks and the average among all of the networks.

In general, all four schemes have quite similar performance within each network and across all networks. During the NN training, all four schemes had similar performance. In some cases scheme R1 had slightly better performance (i.e. ~2% higher $R^2$). For these reasons, and to reduce complexity of the downscaling algorithm, our final downscaling algorithm is scheme R1.

Our downscaling scheme always has equal or better performance compared to the 9km product across individual networks. Since the 9km product is the input to our downscaling, this shows that in terms of temporal correlation and ubRMSE our downscaling algorithm either improves the 9km product or is similar in terms of accuracy. In comparison to the 36km product, our downscaling algorithm's performance follows the 9km product performance.

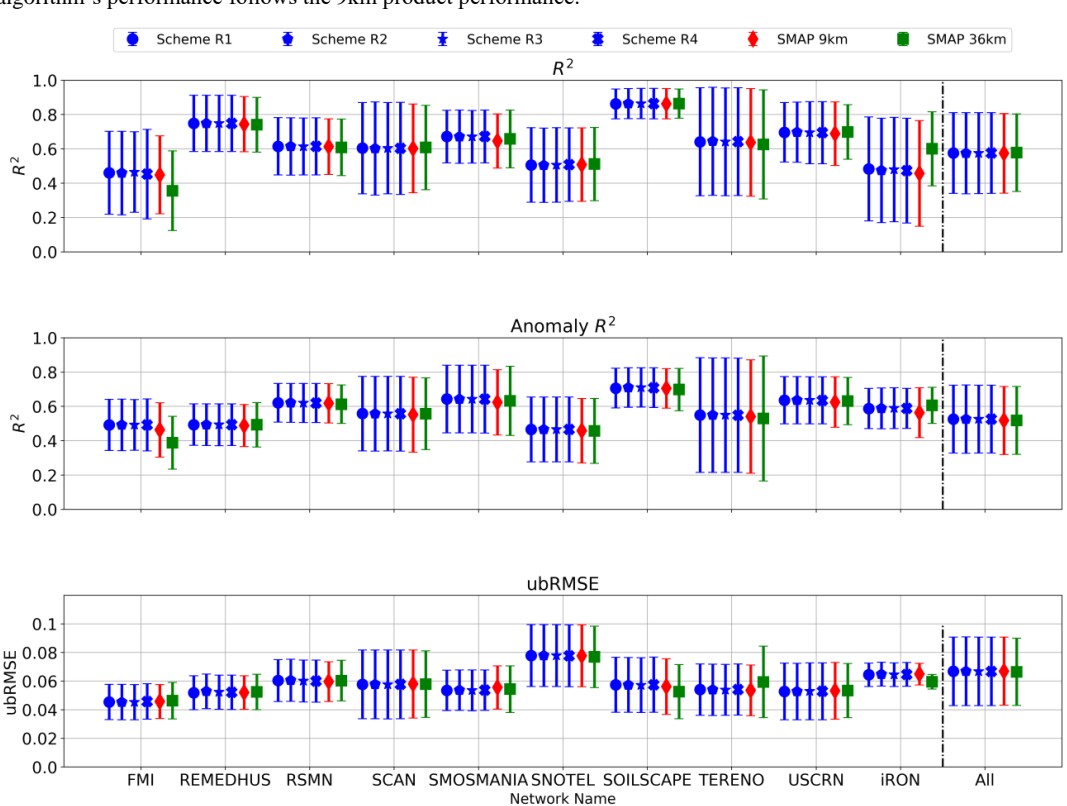

**Figure 8-** $R^2$ **(top), anomaly** $R^2$ **(middle) and ubRMSE (bottom) of each of the downscaling schemes and SMAP 9km and 36km products across 10 different networks of ISMN. Error bars show one standard deviation of each metric around the mean value. Average values across all sites is presented on the right of each panel.**




## 5 Conclusions

In this study, we have developed a statistical disaggregation algorithm using neural networks to downscale soil moisture observations from SMAP to a spatial resolution of 2.25km. We use the two level-3 soil moisture estimates from SMAP at 36km and 9km spatial resolutions to train our downscaling algorithm. Assuming the relationship is consistent between scales, we apply

the downscaling algorithm to 9km soil moisture estimates from SMAP and disaggregate those to 2.25km. Our downscaling algorithm uses only mean monthly NDVI estimates as ancillary data at both the coarse and fine spatial scales to estimate the heterogeneity of soil moisture at sub-pixel level. Our investigation shows that topographic index and variability of NDVI within the coarse scale pixel do not provide additional useful information on the spatial heterogeneity of soil moisture for the downscaling and are thus omitted in the final product. This is the first study to estimate soil moisture at global scale at very fine spatial resolution

(~2km). This new product can be used in a range of other studies and applications including land surface – atmosphere interaction modeling, evapotranspiration modeling and agricultural management. However, in some parts of the world that farm sizes are smaller than the resolution of this product, further disaggregation of soil moisture estimates is needed. Moreover, we will extend these estimates to near real time as long as SMAP soil moisture estimates are available.

Our evaluation shows that the downscaling algorithm has high accuracy in terms of temporal correlation, anomaly correlation and

ubRMSE when compared to in situ soil moisture estimates from ISMN. The performance of the downscaled soil moisture estimates is equal or better than that of the SMAP 9km estimates. Moreover, averaging the estimates at 2.25km within each 9km pixel shows that our downscaling algorithm has high accuracy in preserving the water balance (less than 5% error, within the SMAP mission requirements).

Use of NDVI as an ancillary measurement to disaggregate soil moisture builds on the assumption that there is a moderate vegetation

cover in the pixel of interest. Therefore, this lowers the quality of the downscaling algorithm in bare soil or sparsely vegetated regions. Moreover, NDVI estimates tend to saturate in highly dense vegetated regions such as the tropical forests which results in limited ability of the downscaling algorithm to resolve the heterogeneity of the soil moisture in those regions.

This study shows the potential of using neural networks with a large number of training data to develop a downscaling algorithm for soil moisture. Data generated using this algorithm can be used to provide an improved understanding of the dynamics of soil

moisture and land surface atmosphere interactions at global scale.

### Acknowledgements

The authors would like to acknowledge funding from NASA grant NNX15AB30G. The MYD13A3 data product was retrieved from the online Data Pool, courtesy of the NASA Land Processes Distributed Active Archive Center (LP DAAC), USGS/Earth Resources Observation and Science (EROS) Center, Sioux Falls, South Dakota, https://lpdaac.usgs.gov/data_access/data_pool.

**Data Availability**

Downscaled soil moisture estimates from this study are publicly available, and we intend to extend the temporal coverage of the data periodically. Please contact the corresponding author to access the data: Pierre Gentine (pg2328@columbia.edu)

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
