# Peer review of "Global Downscaling of Remotely-Sensed Soil Moisture using Neural Networks"

_Hydrology and Earth System Sciences, 2017_

## Referee Comment (RC1) · Anonymous Referee #1 · 15 Feb 2018

The manuscript presents a new global soil moisture product provided at an unprecedented spatial resolution of 2.25 km. It is built from a neural network (NN) and data comprised of SMAP 36 km resolution level 3 soil moisture, an enhanced soil moisture product derived from 36 km SMAP observations and posted on a 9 km resolution grid (Chan et al. 2017) and MODIS NDVI data aggregated at various resolutions between 2.25 and 45 km. The authors have also tested the inclusion of a topographic index at the target downscaling resolution. The approach is evaluated by analyzing global soil moisture maps and by comparing downscaled soil moisture estimates against in situ data from the international soil moisture network (ISMN).

A global soil moisture product at 2.25 km resolution is of high interest to the hydrological and Earth system science community. I also find that the comparison of the

[Figure]

NN method with simpler methods (linear interpolation, and the null-hypothesis i.e. no disaggregation) is quite positive as well. In fact, my comments mainly concern the underlying assumptions of the approach (comments #1 and 2) and the evaluation of the downscaled data set (#3).

1) On the use of the 9 km resolution soil moisture product. The basis for the proposed approach is to calibrate a relationship between 36 km (SPL3SMP) and 9 km (SPL3SMP_E) resolution soil moisture products, and then to apply it at 9 km resolution to derive the 2.25 km soil moisture. The point is that the actual spatial resolution of SPM3SMP_E (the so-called "9 km resolution product") is 33 km while it is resampled at 9 km resolution (Chan et al. 2017). The 33 km resolution is so close to the original 36 km resolution SMAP level 3 data that one may wonder how a relationship derived from 36 and 33 km resolution data can be valid between 9 km and 2.25 km resolutions. At the very least, I recommend a sensitive analysis to assess the impact on the results of a coarser spatial resolution (33 km instead of 9 km) for training.

2) The NN is trained and run using NDVI data as auxiliary information about the sub-pixel soil moisture variability. Some limitations related to the soil moisture-NDVI relationship are mentioned in the conclusion (such as presence of vegetation, saturation effects). However, I think that the discussion should be deepened. It is true that NDVI and topography are variables available at global scale, but they are not the only factors explaining the soil moisture variability. In addition, the soil moisture-NDVI relationship established at the monthly time scale (phenological time scale) may not be valid at the daily time scale, at which SMAP observes the Earth and the observed surface soil moisture evolves. For clarity, the assumptions underlying the implementation of the NN using NDVI data should be better highlighted in the manuscript.

3) Evaluation of the NN output: Line 22 page 9: "NN is appropriately explaining the spatial variability of soil moisture using NDVI as ancillary data". Line 14 page 13: "our evaluation shows that the downscaling algorithm has high accuracy in terms of temporal correlation, anomaly correlation and ubRMSE when compared to in situ soil

moisture estimates from ISMN". It is difficult to assess the quality of the downscaled soil moisture at fine scale using global maps. Global maps convey the message that the high-resolution product is global, but some fine scale assessment is missing. Evaluation of the results over focus (perhaps instrumented) areas would be very useful. Regarding the temporal aspect, validation using 2-year averages does not allow for assessing the relevance of soil moisture-NDVI relationships at the temporal scale of SMAP observations/surface soil moisture dynamics. In addition, I do not think that the comparison with in situ measurements shows "that the downscaling algorithm has high accuracy". I would soften this point of view as results are very similar for all products (from 2.25 km to 36 km, see Figure 8). Even though the downscaling method does not degrade low resolution information, the improvement is hard to detect. The authors mention that "accuracy is better than or equal to the SMAP 9 km soil moisture estimates". I take them at their word, but from Figure 8 it seems that the original 36 km product be more accurate at several stations. More explanations are needed to clarify the improvement provided at 2.25 km and at which temporal scale.

Specific points:

a) I may have missed something but the two statements re-written below look contradictory: - Page 9, line 11: "Moreover, the latitudinal average plots (on the right side of each panel of Figure 5) show that at higher spatial resolutions there is more spatial heterogeneity. The latitudinal average for the 36 km product is much smoother than the 2.25 km one." - page 9, line 18 : "For comparison, we also calculate CV for the 9km soil moisture estimates from SMAP at the 36km grid (Figure 6 bottom panel). The two panels in Figure 6 have different range of CV which is expected given the difference in their spatial scales." Since aggregation tends to reduce variabilities, one would expect an increase in the spatial variability at higher spatial resolution. However, the CV is divided by about 5 at 9 km resolution compared to the CV at 36 km resolution. Could the authors comment on their seemingly opposite findings ?

b) Line 13 page 2: "Some of them use linear relationships (i.e. projection) to define

the impact of spatial heterogeneity using ancillary data, typically in combination with a radiative-transfer model to relate surface temperature and soil moisture (Colliander et al., 2017a; Merlin et al. 2005, 2008a, 2008b, 2008c)." I noted two errors in this sentence: 1) physical models that relate surface temperature and soil moisture are energy balance models (not radiative transfer models) and 2) the projection technique used in Merlin et al. 2005, 2008a does not implement linear relationships, but a non-linear energy balance model.

c) Line 15 page 2: "A major issue is that surface temperature at finer spatial scales from satellites cannot be estimated under cloudy conditions". Agree and I would add another essential limitation that the surface temperature cannot be used as a signature of soil moisture in energy-limited conditions.

d) Line 21 page 4: "we assume that the scaling relationship between 36 and 9 km soil moisture estimates is the same as the scaling relationship between 9 km and 2.25 km resolution. To the best of our knowledge, this is the first time that the assumption of similar scaling relationship is used to downscale soil moisture". I would like to mention that the same scaling relationship has already been used to downscale soil moisture from 40 km to 1 km and from 1 km and 100 m in Merlin et al. 2009 and Merlin et al. 2013. Merlin, O., Al Bitar, A., Walker, J. P., & Kerr, Y. (2009). A sequential model for disaggregating near-surface soil moisture observations using multi-resolution thermal sensors. Remote Sensing of Environment, 113(10), 2275-2284. Merlin, O., Escori-huela, M. J., Mayoral, M. A., Hagolle, O., Al Bitar, A., & Kerr, Y. (2013). Self-calibrated evaporation-based disaggregation of SMOS soil moisture: An evaluation study at 3 km and 100 m resolution in Catalunya, Spain. Remote sensing of environment, 130, 25-38.

---

## Referee Comment (RC2) · Anonymous Referee #2 · 16 Apr 2018

The authors have downscaled global satellite-based soil moisture observations to 2.25km spatial resolution using neural networks. The manuscript is well written, and the methodology is sound. The high-resolution products will be useful for global hydrological and climate studies. However, there are a few issues in this manuscript, hence I suggest some major revisions.

My major concerns are:

1. The training and retrieval in the NN algorithm are based on the hypothesis that it shares the same "relationship" between inputs and outputs to downscale soil moisture from 36km to 9km and to downscale soil moisture from 9km to 2.25km. The author should provide more information about how the SPL3SMP_E product is enhanced from the SPL3SMP product. As the author mentioned, the native resolution of SPL3SMP_E

is actually about 33km. So what is the relationship among 36km, 33km, and 9km?

2. It is a little confusing to add standard deviation of NDVI in the downscale scheme. All the schemes have included NDVI at the higher resolution (9 km for training and 2.25km for retrieval). What additional improvement will the standard deviation of NDVI provide? As the author stated, "This estimate provides a proxy of the heterogeneity within the coarse scale grid". Hasn't the higher-resolution NDVI already provided information about "sub-pixel heterogeneity"? This makes the conclusions somehow self-contradictory. The key of the downscaling algorithm is the higher-resolution NDVI "as an ancillary data to quantify sub-pixel heterogeneity of soil moisture". However, the similar performance of the four schemes suggests that "variability of NDVI within the coarse scale pixel does not provide additional useful information on the spatial heterogeneity of soil moisture for the downscaling".

Specific points:

P2, L29: "Our final product is soil moisture estimates at 2.25km spatial resolution with full global coverage every 2-3 days ... " Is it possible to explain why the final product is downscaled at 2.25km spatial resolution here?

P4, L8: What are the temporal resolution of the in situ soil moisture observations?

P4, L13: 20 measurements at one station?

P4, L14: What is the resolution of the "pixel" here? Does one pixel only contain one station?

P7, Table 1: Are they inputs only for training? It would be better to include the information about retrieval.

P9, L8: The maps look pretty similar, can you tell any difference between them, or which one is better?

P9, L12-13: "... at higher spatial resolutions there is more spatial heterogeneity. The

latitudinal average for the 36 km product is much smoother than the 2.25 km one." What is the purpose of this? Does more spatial heterogeneity indicate better quality?

P11, Figure 7: Any explanation for the dry bias over the arid area?

P12, L2: How are the metrics calculated? calculate correlation temporally, then average among the pixels/stations? About the error bars on Figure 8, are they standard deviation among pixels or anything else?

P12, L4: For Figure S6, It would be better to show the 10 networks with different colors or symbols on the map... Also, this map should have been mentioned in section 2.4.

P12, Figure 8: It is really hard to tell from the figure if there is any better performance of the downscaled products than the 9km SMAP product.

P12, Figure 8: Any idea why there is poor performance over some network (e.g., SNO-TEL) but good performance over some other networks (e.g. SOILSCAPE). Why does SMAP 36km have the best agreement with iRON?

P13, L7-9: This conclusion might only be true among the different NN downscaling schemes.

P13, L20: "... this lowers the quality of the downscaling algorithm in bare soil or sparsely vegetated." Do any of the results support this statement? According to Figure 4, there is very high correlation coefficient over those regions.

---

## Author Comment (AC2) · 20 May 2018

*We thank Referee #2 for their comments. Here, we respond to the general comments and specific points.*

**General Comments:**

The authors have downscaled global satellite-based soil moisture observations to 2.25km spatial resolution using neural networks. The manuscript is well written, and the methodology is sound. The high-resolution products will be useful for global hydro- logical and climate studies. However, there are a few issues in this manuscript, hence I suggest some major revisions.

My major concerns are:

[Figure]

1. The training and retrieval in the NN algorithm are based on the hypothesis that it shares the same "relationship" between inputs and outputs to downscale soil moisture from 36km to 9km and to downscale soil moisture from 9km to 2.25km. The author should provide more information about how the SPL3SMP_E product is enhanced from the SPL3SMP product. As the author mentioned, the native resolution of SPL3SMP_E is actually about 33km. So what is the relationship among 36km, 33km, and 9km?

*Response/Action: In this study, we made the assumption that the scaling relationship from the 36km product to the 9km product can be used to estimate soil moisture at 2.25km. We have independently validated the soil moisture values at the 2.25km scale; therefore, we believe this is a valid assumption.*

*We agree with the difference between native resolution and grid spacing of the SPM3SMP_E product. The 9km product applies the Backus-Gilbert (BG) optimal interpolation technique to the oversampled radiometer measurements to estimate brightness temperature (TB) at each point, which is the center of each 9km pixel. Meanwhile, the native resolution of the data which is the spatial extent projected on earth surface by the 3-dB beamwidth of the radiometer is very close to the 36km product.*

*Contrary to this approach, the SMAP 36km product uses the antenna pattern information to estimate TB from the original radiometer measurements, and does not take into account the oversampling effect. This results in a coarser grid resolution data.*

2. It is a little confusing to add standard deviation of NDVI in the downscale scheme. All the schemes have included NDVI at the higher resolution (9 km for training and 2.25km for retrieval). What additional improvement will the standard deviation of NDVI provide? As the author stated, "This estimate provides a proxy of the heterogeneity within the coarse scale grid". Hasn't the higher-resolution NDVI already provided information about "sub-pixel heterogeneity"? This makes

the conclusions somehow self-contradictory. The key of the downscaling algorithm is the higher-resolution NDVI "as an ancillary data to quantify sub-pixel heterogeneity of soil moisture". However, the similar performance of the four schemes suggests that "variability of NDVI within the coarse scale pixel does not provide additional useful information on the spatial heterogeneity of soil moisture for the downscaling".

***Response/Action:*** *We would like to highlight that the final algorithm (scheme R1) which is used to retrieve soil moisture at the downscaled 2.25km scale does not include standard deviation of NDVI (NDVI_std). The reason we included NDVI_std in schemes R2 and R4 was to check if an explicit input of the variability of the NDVI (reflecting within watershed variations) within each coarse scale pixel (36km during training and 9km during estimation) would increase the accuracy of the soil moisture retrieval at the finer scale. NN algorithms apply a combination of linear operations to the inputs, but not necessarily power operations that might inform the network of the variance/std of the inputs. So we made an assumption, added this variability as an input, tested the results, and reported that it does not improve the accuracy of estimation. Therefore, to reduce complexity of the model we did not include NDVI_std (or TI for the same reason) in the final algorithm. We believe reporting this analysis is informative for the readers who might be interested in expanding our approach and building new downscaling algorithms, especially to account for watershed variations in soil moisture.*

**Specific Comments:**

- P2, L29: "Our final product is soil moisture estimates at 2.25km spatial resolution with full global coverage every 2-3 days ... " Is it possible to explain why the final product is downscaled at 2.25km spatial resolution here?

***Response/Action:*** *We will add the following sentence in this paragraph to explain the reason for the choice of 2.25km data:*

*"The 2.25km spatial resolution is chosen since we use SMAP 36km and 9km soil moisture products for training and developing the scaling relationship. This is used to estimate soil moisture at the finer 2.25km resolution which is $\frac{1}{4}$ of 9km."*

- P4, L8: What are the temporal resolution of the in situ soil moisture observations?

***Response/Action:*** *They are hourly measurements, but we have only used the measurement at 6am local time which is concurrent with SMAP measurement.*

- P4, L13: 20 measurements at one station?

***Response/Action:*** *We apologize for the confusion, we meant 20 measurements in time at each station. We have excluded any station that has less than 20 measurements during the study period because temporal statistics won't be meaningful. We will revise the sentence in the revised manuscript.*

- P4, L14: What is the resolution of the "pixel" here? Does one pixel only contain one station?

***Response/Action:*** *We apologize for the mistake. We meant stations and not pixels. The numbers for each network refer to the number of stations used from each. We will correct this in the revised manuscript.*

- P7, Table 1: Are they inputs only for training? It would be better to include the information about retrieval.

***Response/Action:*** *The information for training and retrieval are the same but their spatial resolution differ. We will change the table in the revised manuscript to reflect the specifications of the input data for retrieval.*

- P9, L8: The maps look pretty similar, can you tell any difference between them, or which one is better?

*Response/Action: We expect them to be similar at this scale. However, there are differences in regions such as the Sahara and regions with high soil moisture gradient (near Congo basin, and the Amazon). Other differences are in the latitudinal averages and pixels that do not have soil moisture estimate (such as major rivers). Due to the suggestion of referee 1, we will include soil moisture maps from focus regions in the revised manuscript that will depict differences between the products.*

- P9, L12-13: "...at higher spatial resolutions there is more spatial heterogeneity. The latitudinal average for the 36 km product is much smoother than the 2.25 km one." What is the purpose of this? Does more spatial heterogeneity indicate better quality?

*Response/Action: It is a measure of quality relative to the spatial resolution. Soil moisture at higher spatial resolution is more informative for applications at small scale. This comparison shows that the soil moisture estimate at 2.25km has indeed more variability, and is not just an interpolated version of the coarser resolution product.*

- P11, Figure 7: Any explanation for the dry bias over the arid area?

*Response/Action: We had briefly noted this in the conclusion section (P13, L19-22):*

*"Use of NDVI as an ancillary measurement to disaggregate soil moisture builds on the assumption that there is a moderate vegetation cover in the pixel of interest. Therefore, this lowers the quality of the downscaling algorithm in bare soil or sparsely vegetated regions. Moreover, NDVI estimates tend to saturate in highly dense vegetated regions such as the tropical forests which results in limited ability of the downscaling algorithm to resolve the heterogeneity of the soil moisture in those regions."*

*We will revise this paragraph as following:*

*"Use of NDVI as an ancillary measurement to disaggregate soil moisture builds on the assumption that there is a moderate vegetation cover in the pixel of interest. Therefore, this lowers the quality of the downscaling algorithm in bare soil or sparsely vegetated regions. Figure 7 shows a dry bias in arid regions which is an indication of lack of NDVI – soil moisture relationship. Moreover, NDVI estimates tend to saturate in highly dense vegetated regions such as the tropical forests which results in limited ability of the downscaling algorithm to resolve the heterogeneity of the soil moisture in those regions."*

- P12, L2: How are the metrics calculated? calculate correlation temporally, then average among the pixels/stations? About the error bars on Figure 8, are they standard deviation among pixels or anything else?

*Response/Action: For each pixel that has in situ observations one set of metrics is calculated. If more than one station from a network fall within a pixel, their observations are averaged before calculating the metrics. Finally, metrics from each network are summarized in Figure 8 with mean and standard deviation among pixels.*

- P12, L4: For Figure S6, It would be better to show the 10 networks with different colors or symbols on the map... Also, this map should have been mentioned in section 2.4.

*Response/Action: We will change the figure as requested by the referee and reference it in Section 2.4.*

- P12, Figure 8: It is really hard to tell from the figure if there is any better performance of the downscaled products than the 9km SMAP product.

*Response/Action: This figure summarizes the performance of different products with respect to temporal dynamics metrics. Since the downscaled product is*

*derived from the 9km product, we do not expect to see significant improvement in temporal dynamics of soil moisture estimates. Rather we check to verify that the increased spatial resolution in the downscaled product does not decrease the temporal performance.*

- P12, Figure 8: Any idea why there is poor performance over some network (e.g., SNOTEL) but good performance over some other networks (e.g. SOILSCAPE). Why does SMAP 36km have the best agreement with iRON?

***Response/Action:** While this assessment is beyond the scope of this manuscript, we would like to note that while iRON has the best agreement with SMAP data with respect to ubRMSE it does not have the best correlation. While SOILSCAPE has the best correlation, it has an average ubRMSE metric. More detailed assessment of SMAP retrievals across different ground truth stations is provided in:*

*Chan, S. K. et al. (2016) 'Assessment of the SMAP Passive Soil Moisture Product', IEEE Transactions on Geoscience and Remote Sensing, 54(8), pp. 4994–5007. doi: 10.1109/TGRS.2016.2561938.*

- P13, L7-9: This conclusion might only be true among the different NN downscaling schemes.

***Response/Action:** We agree with the referee, and will revise this statement as following in the revised manuscript:*

*"Our investigation shows that topographic index and variability of NDVI within the coarse scale pixel do not provide additional useful information on the spatial heterogeneity of soil moisture for the downscaling using our proposed NN technique and are thus omitted in the final product."*

- P13, L20: ". . . this lowers the quality of the downscaling algorithm in bare soil or

sparsely vegetated." Do any of the results support this statement? According to Figure 4, there is very high correlation coefficient over those regions.

***Response/Action:*** *Lower quality is with respect to both correlation and bias. Figure 4 shows that those regions have high correlation, but Figure 7 shows the same regions have a negative bias in soil moisture estimates. We will revise the sentence as following:*

*"Therefore, this lowers the quality of the downscaling algorithm in bare soil or sparsely vegetated regions as indicated by negative bias in arid regions in Figure 7.*

---

## Author Response (AR1)

Response to Editor's Comments
**Global Downscaling of Remotely-Sensed Soil Moisture using Neural Networks**
**(Manuscript # hess-2017-680)**

Thank you for providing a detailed and pointed response to the reviewers' comments. Overall I am satisfied with the response however I wonder if more can be done to address reviewer #1's comment #1. E.g. can you not generate a 2.25 Km version directly from the 36 Km version and compare it with the version of 2.25 Km that is generated using 9km? I think the reviewer has raised an important point about the uncertainty in the downscaling relationship and it should be satisfactorily addressed.

*Response/Action: We appreciate Editor's suggestion, and acknowledge the importance of the scaling assumption. However, as we mentioned in the response to referee #1's comment there is no way to generate a downscaled soil moisture estimate at 2.25km directly from the 36km product. The NN approach needs to be trained on a similar scaled product (in this case 36km to 9km), and then be used with the 9km product as input to generate the 2.25 km product. If we want to generate the 2.25km product directly from 36km product, we need a training datasets with 1/8 of scale between the spatial resolution of input and output which is not available.*

Response to Referee #1 Comments
**Global Downscaling of Remotely-Sensed Soil Moisture using Neural Networks
(Manuscript # hess-2017-680)**

*We thank Referee #1 for their positive comments. We respond to the general comments and specific points in the following.*

**General Comments:**

- The manuscript presents a new global soil moisture product provided at an unprecedented spatial resolution of 2.25 km. It is built from a neural network (NN) and data comprised of SMAP 36 km resolution level 3 soil moisture, an enhanced soil moisture product derived from 36 km SMAP observations and posted on a 9 km resolution grid (Chan et al. 2017) and MODIS NDVI data aggregated at various resolutions between 2.25 and 45 km. The authors have also tested the inclusion of a topographic index at the target downscaling resolution. The approach is evaluated by analyzing global soil moisture maps and by comparing downscaled soil moisture estimates against in situ data from the international soil moisture network (ISMN).
A global soil moisture product at 2.25 km resolution is of high interest to the hydrological and Earth system science community. I also find that the comparison of the NN method with simpler methods (linear interpolation, and the null-hypothesis i.e. no disaggregation) is quite positive as well. In fact, my comments mainly concern the underlying assumptions of the approach (comments #1 and 2) and the evaluation of the downscaled data set (#3).

1) On the use of the 9 km resolution soil moisture product. The basis for the proposed approach is to calibrate a relationship between 36 km (SPL3SMP) and 9 km (SPL3SMP_E) resolution soil moisture products, and then to apply it at 9 km resolution to derive the 2.25 km soil moisture. The point is that the actual spatial resolution of SPM3SMP_E (the so-called "9 km resolution product") is 33 km while it is resampled at 9 km resolution (Chan et al. 2017). The 33 km resolution is so close to the original 36 km resolution SMAP level 3 data that one may wonder how a relationship derived from 36 and 33 km resolution data can be valid between 9 km and 2.25 km resolutions. At the very least, I recommend a sensitive analysis to assess the impact on the results of a coarser spatial resolution (33 km instead of 9 km) for training.

*Response/Action: We agree with the difference between native resolution and grid spacing of the SPM3SMP_E product. Indeed, we had included this in the product description (Section 2.1) of the original submission. However, the methodology to generate gridded brightness temperature (TB) (and subsequently soil moisture) from original observations of the SMAP antenna is different in the 36km and 9km product. The 9km product applies the Backus-Gilbert (BG) optimal interpolation technique to the oversampled radiometer measurements to estimate TB at each point, which is the center of each 9km pixel. Meanwhile, the native*

*resolution of the data which is the spatial extent projected on earth surface by the 3-dB beamwidth of the radiometer is very close to the 36km product.*

*Contrary to this approach, the SMAP 36km product uses the antenna pattern information to estimate TB from the original radiometer measurements, and does not take into account the oversampling effect. This results in a coarser grid resolution data.*

*Unfortunately, we cannot conduct a sensitivity analysis since the main assumption is that the downscaling has the same scaling ratio during training and estimation (from 36km to 9km, and from 9km to 2.25km). Moreover, the enhanced soil moisture product is already on a 9km grid. If we want to upscale that to 33km it adds some uncertainty to the sensitivity analysis (there is no soil moisture product from SMAP on 33km grid).*

*In this study, we made the assumption that the scaling relationship from the 36km product to the 9km product can be used to estimate soil moisture at 2.25km. We have independently validated the soil moisture values at the 2.25km scale; therefore, we believe this is a reasonable assumption.*

*Unfortunately, we cannot test such relationship between the different scales (we would need indeed 2.25, 9 and 36km data and even more spatial dependence to test the scaling relationship across scales). This is currently impossible nonetheless but we hope that future high resolution mission could allow such test of the scaling relationship to inform such retrieval which could be adapted in the future to include some scale dependence.*

2) The NN is trained and run using NDVI data as auxiliary information about the sub-pixel soil moisture variability. Some limitations related to the soil moisture-NDVI relationship are mentioned in the conclusion (such as presence of vegetation, saturation effects). However, I think that the discussion should be deepened. It is true that NDVI and topography are variables available at global scale, but they are not the only factors explaining the soil moisture variability. In addition, the soil moisture-NDVI relationship established at the monthly time scale (phenological time scale) may not be valid at the daily time scale, at which SMAP observes the Earth and the observed surface soil moisture evolves. For clarity, the assumptions underlying the implementation of the NN using NDVI data should be better highlighted in the manuscript.

*Response/Action: We agree with the referee on the limitations of NDVI – SM relationship. The assumption we make in using NDVI as an auxiliary information to predict variability of soil moisture is that within a neighborhood of a specific pixel NDVI of that pixel 'relative' to the NDVI of the neighboring pixels is an indicator of the wetness or dryness of that pixel with respect to its neighboring pixels. This is different than using the 'absolute' value of NDVI for soil moisture prediction.*
*However, to clarify this point, we added the following paragraph in the conclusion section (P. 16, L. 3-7):*
*"In this study, we use the relative value of NDVI (in a given pixel with respect to the neighboring pixels) as an auxiliary information to predict spatial variability of*

*soil moisture in each coarse-scale pixel. While the relationship between soil moisture and NDVI at phenological time scales may not be valid at the temporal scale of SMAP observations (couple of days), our assumption builds on the relative value of NDVI within a small region and not the absolute value. Therefore, it is reasonable to use NDVI as a predictor in this case"*

3) Evaluation of the NN output: Line 22 page 9: "NN is appropriately explaining the spatial variability of soil moisture using NDVI as ancillary data". Line 14 page 13: "our evaluation shows that the downscaling algorithm has high accuracy in terms of temporal correlation, anomaly correlation and ubRMSE when compared to in situ soil moisture estimates from ISMN". It is difficult to assess the quality of the downscaled soil moisture at fine scale using global maps. Global maps convey the message that the high-resolution product is global, but some fine scale assessment is missing. Evaluation of the results over focus (perhaps instrumented) areas would be very useful. Regarding the temporal aspect, validation using 2-year averages does not allow for assessing the relevance of soil moisture-NDVI relationships at the temporal scale of SMAP observations/surface soil moisture dynamics. In addition, I do not think that the comparison with in situ measurements shows "that the downscaling algorithm has high accuracy". I would soften this point of view as results are very similar for all products (from 2.25 km to 36 km, see Figure 8). Even though the downscaling method does not degrade low resolution information, the improvement is hard to detect. The authors mention that "accuracy is better than or equal to the SMAP 9 km soil moisture estimates". I take them at their word, but from Figure 8 it seems that the original 36 km product be more accurate at several stations. More explanations are needed to clarify the improvement provided at 2.25 km and at which temporal scale.

*Response/Action: We agree with referee's comment that conducting evaluations over focus regions would be informative. We added this to the revised manuscript. The goal of the comparison with in-situ observations is to assess temporal accuracy of the downscaled product. Since the input data to the soil moisture estimates at 2.25km is the 9km product, we did not expect to get significantly higher temporal accuracy with respect to the 9km product. The value of the 2.25km product is an enhanced spatial resolution, while having the same temporal accuracy. We revised the statements in the results section highlighted by the referee to clarify this.*

**Specific points:**

a) I may have missed something but the two statements re-written below look contradictory: - Page 9, line 11: "Moreover, the latitudinal average plots (on the right side of each panel of Figure 5) show that at higher spatial resolutions there is more spatial heterogeneity. The latitudinal average for the 36 km product is much

smoother than the 2.25 km one." - page 9, line 18 : "For comparison, we also calculate CV for the 9km soil moisture estimates from SMAP at the 36km grid (Figure 6 bottom panel). The two panels in Figure 6 have different range of CV which is expected given the difference in their spatial scales." Since aggregation tends to reduce variabilities, one would expect an increase in the spatial variability at higher spatial resolution. However, the CV is divided by about 5 at 9 km resolution compared to the CV at 36 km resolution. Could the authors comment on their seemingly opposite findings ?

*Response/Action: Indeed, these two statements are not contradictory. The first one referring to Figure 5 is based on the fact that at higher spatial resolution we are seeing more variability. Mainly, the latitudinal plots capture the smaller changes in soil moisture that were not observable with the coarse resolution product.*
*On the other hand, the CV plots in Figure 6 show the variability of soil moisture within a 9km pixel and within a 36km pixel. Within a 9km pixel the variability of soil moisture is smaller than that of a 36km pixel, since the surface is less heterogeneous (this is true for a 9km pixel which is spatially within the 9km pixel). It is true that the 2.25km product should show more heterogeneity compared to the 9km product but this would be true if both 9km and 2.25km product variabilities are compared at the 36km scale. Currently, we are comparing variability of the 2.25km product within the 9km pixel, and variability of the 9km pixel within the 36km pixel. We will add the CV plot of the 2.25km product within each 36km pixel in the revised manuscript to better explain the spatial variability of soil moisture.*

b) Line 13 page 2: "Some of them use linear relationships (i.e. projection) to define the impact of spatial heterogeneity using ancillary data, typically in combination with a radiative-transfer model to relate surface temperature and soil moisture (Colliander et al., 2017a; Merlin et al. 2005, 2008a, 2008b, 2008c)." I noted two errors in this sentence: 1) physical models that relate surface temperature and soil moisture are energy balance models (not radiative transfer models) and 2) the projection technique used in Merlin et al. 2005, 2008a does not implement linear relationships, but a non- linear energy balance model.

*Response/Action: We agree with the referee's comment, and revised this section to correctly summarize the work done in the literature.*

c) Line 15 page 2: "A major issue is that surface temperature at finer spatial scales from satellites cannot be estimated under cloudy conditions". Agree and I would add another essential limitation that the surface temperature cannot be used as a signature of soil moisture in energy-limited conditions.

*Response/Action: We agree with this limitation, and added it to the revised manuscript.*

d) Line 21 page 4: "we assume that the scaling relationship between 36 and 9 km soil moisture estimates is the same as the scaling relationship between 9 km and 2.25 km resolution. To the best of our knowledge, this is the first time that the assumption of similar scaling relationship is used to downscale soil moisture". I would like to mention that the same scaling relationship has already been used to downscale soil moisture from 40 km to 1 km and from 1 km and 100 m in Merline tal. 2009 and Merlinetal. 2013.

Merlin, O., Al Bitar, A., Walker, J. P., & Kerr, Y. (2009). A sequential model for disaggregating near-surface soil moisture observations using multi-resolution thermal sensors. Remote Sensing of Environment, 113(10), 2275-2284.

Merlin, O., Escori- huela, M. J., Mayoral, M. A., Hagolle, O., Al Bitar, A., & Kerr, Y. (2013). Self-calibrated evaporation-based disaggregation of SMOS soil moisture: An evaluation study at 3 km and 100 m resolution in Catalunya, Spain. Remote sensing of environment, 130, 25-38.

*Response/Action: We acknowledge new references provided by the referee. However, neither of the studies make the same assumption as we did between different scales of soil moisture. Both of the studies developed a sequential disaggregation in which the first step uses MODIS data to disaggregate soil moisture from SMOS resolution to MODIS resolution, next the MODIS-disaggregated soil moisture is further downscaled using ASTER and/or Landsat data. Unlike these two studies, we did not develop a sequential downscaling approach. The 9km product used in our study is estimated by different technique form direct brightness observations of the satellite. We find a relationship between the soil moisture estimates at 36km and 9km using some ancillary data, and then apply this relationship to the 9km product to estimate soil moisture at 2.25km. Those references have in any case been added to the list of references.*

Response to Referee #2 Comments
**Global Downscaling of Remotely-Sensed Soil Moisture using Neural Networks**
**(Manuscript # hess-2017-680)**

*We thank Referee #2 for their comments. Here, we respond to the general comments and specific points:*

**General Comments:**

The authors have downscaled global satellite-based soil moisture observations to 2.25km spatial resolution using neural networks. The manuscript is well written, and the methodology is sound. The high-resolution products will be useful for global hydro- logical and climate studies. However, there are a few issues in this manuscript, hence I suggest some major revisions.
My major concerns are:

1. The training and retrieval in the NN algorithm are based on the hypothesis that it shares the same "relationship" between inputs and outputs to downscale soil moisture from 36km to 9km and to downscale soil moisture from 9km to 2.25km. The author should provide more information about how the SPL3SMP_E product is enhanced from the SPL3SMP product. As the author mentioned, the native resolution of SPL3SMP_E is actually about 33km. So what is the relationship among 36km, 33km, and 9km?

*Response/Action: In this study, we made the assumption that the scaling relationship from the 36km product to the 9km product can be used to estimate soil moisture at 2.25km. We have independently validated the soil moisture values at the 2.25km scale; therefore, we believe this is a valid assumption.*
*We agree with the difference between native resolution and grid spacing of the SPM3SMP_E product. The 9km product applies the Backus-Gilbert (BG) optimal interpolation technique to the oversampled radiometer measurements to estimate brightness temperature (TB) at each point, which is the center of each 9km pixel. Meanwhile, the native resolution of the data which is the spatial extent projected on earth surface by the 3-dB beamwidth of the radiometer is very close to the 36km product.*
*Contrary to this approach, the SMAP 36km product uses the antenna pattern information to estimate TB from the original radiometer measurements, and does not take into account the oversampling effect. This results in a coarser grid resolution data.*
*Unfortunately, we cannot test such relationship between the different scales (we would need indeed 2.25. 9 and 36km data and even more spatial dependence to test the scaling relationship across scales). This is currently impossible nonetheless but we hope that future high resolution mission could allow such test of the scaling relationship to inform such retrieval which could be adapted in the future to include some scale dependence.*

2. It is a little confusing to add standard deviation of NDVI in the downscale scheme. All the schemes have included NDVI at the higher resolution (9 km for training and 2.25km for retrieval). What additional improvement will the standard deviation of NDVI provide? As the author stated, "This estimate provides a proxy of the heterogeneity within the coarse scale grid". Hasn't the higher-resolution NDVI already provided information about "sub-pixel heterogeneity"? This makes the conclusions somehow self-contradictory. The key of the downscaling algorithm is the higher-resolution NDVI "as an ancillary data to quantify sub-pixel heterogeneity of soil moisture". However, the similar performance of the four schemes suggests that "variability of NDVI within the coarse scale pixel does not provide additional useful information on the spatial heterogeneity of soil moisture for the downscaling".

*Response/Action: We would like to highlight that the final algorithm (scheme R1) which is used to generate soil moisture at the downscaled 2.25km scale does not include standard deviation of NDVI (NDVI_std). The reason we included NDVI_std in scheme R2 and R4 was to check if an explicit input of the variability of the NDVI within each coarse scale pixel (36km during training and 9km during estimation) would increase the accuracy of the soil moisture estimation at the finer scale. NN algorithms apply a combination of linear operations to the inputs, but not necessarily power operations that might inform the network of the variance/std of the inputs. So we made an assumption, added this variability as an input, tested the results and this hypothesis, and reported that it does not improve the accuracy of estimation. Therefore, to reduce complexity of the model we did not include NDVI_std (or TI for the same reason) in the final algorithm. We believe reporting this analysis is informative for the readers who might be interested in expanding our approach and building new downscaling algorithms.*

**Specific points:**

- P2, L29: "Our final product is soil moisture estimates at 2.25km spatial resolution with full global coverage every 2-3 days ... " Is it possible to explain why the final product is downscaled at 2.25km spatial resolution here?
  *Response/Action: We added the following sentence in this paragraph to explain the reason for choice of 2.25km:*
  *"The 2.25km spatial resolution is chosen since we use SMAP 36km and 9km (¼ of 36km) soil moisture products for training, and developing the scaling relationship. This relationship is used to estimate soil moisture at the finer 2.25km resolution which is ¼ of 9km."*

- P4, L8: What are the temporal resolution of the in situ soil moisture observations?

***Response/Action:*** *They are hourly measurements, but we have only used the measurement at 6am local time which is concurrent with SMAP measurement.*

- P4, L13: 20 measurements at one station?
***Response/Action:*** *We apologize for the confusion, we meant 20 measurements in time in each station. We have excluded any station that has less than 20 measurements during the study period because temporal statistics won't be meaningful. We changed the sentence in the revised manuscript.*

- P4, L14: What is the resolution of the "pixel" here? Does one pixel only contain one station?
***Response/Action:*** *We apologize for the mistake. We mean stations not pixels. The numbers for each network refer to the number of stations used from each. We corrected this in the revised manuscript.*

- P7, Table 1: Are they inputs only for training? It would be better to include the information about retrieval.
***Response/Action:*** *The information for training and retrieval are the same but their spatial resolution is different. We changed the table in the revised manuscript to reflect the specifications of the input data for retrieval.*

- P9, L8: The maps look pretty similar, can you tell any difference between them, or which one is better?
***Response/Action:*** *We expect them to be similar at this scale. However, there are differences in regions such as the Sahara and regions with high soil moisture gradient (near Congo basin, and the Amazon). Other differences are in the latitudinal averages and pixels that do not have soil moisture estimate (such as major rivers). Due to the suggestion of referee 1, we included soil moisture maps from focus regions in the revised manuscript that depict differences between the products. (Figure 6)*

- P9, L12-13: ". . . at higher spatial resolutions there is more spatial heterogeneity. The latitudinal average for the 36 km product is much smoother than the 2.25 km one." What is the purpose of this? Does more spatial heterogeneity indicate better quality?
***Response/Action:*** *It is a measure of quality relative to the spatial resolution. Soil moisture at higher spatial resolution is more informative for applications at small scale. This comparison shows that the soil moisture estimate at 2.25km has indeed more variability, and it not an interpolated version of the coarser resolution product.*

- P11, Figure 7: Any explanation for the dry bias over the arid area?
***Response/Action:*** *We had noted this in the conclusion section of the original manuscript (P13, L19-22):*
*"Use of NDVI as an ancillary measurement to disaggregate soil moisture builds on the assumption that there is a moderate vegetation cover in the pixel of interest. Therefore, this lowers the quality of the downscaling algorithm in bare soil or*

*sparsely vegetated regions. Moreover, NDVI estimates tend to saturate in highly dense vegetated regions such as the tropical forests which results in limited ability of the downscaling algorithm to resolve the heterogeneity of the soil moisture in those regions."*

*We will revised this paragraph as following:*
*"Use of NDVI as an ancillary measurement to disaggregate soil moisture builds on the assumption that there is a moderate vegetation cover in the pixel of interest. Therefore, this lowers the quality of the downscaling algorithm in bare soil or sparsely vegetated regions. Figure 7 shows a dry bias in arid regions which is an indication of lack of NDVI – soil moisture relationship. Moreover, NDVI estimates tend to saturate in highly dense vegetated regions such as the tropical forests which results in limited ability of the downscaling algorithm to resolve the heterogeneity of the soil moisture in those regions."*

- P12, L2: How are the metrics calculated? calculate correlation temporally, then average among the pixels/stations? About the error bars on Figure 8, are they standard deviation among pixels or anything else?
  *Response/Action: For each pixel that has in situ observations one set of metrics is calculated. If more than one station from a network fall within a pixel, their observations are averaged before calculating the metrics. Finally, metrics from each station are summarized in Figure 8 with mean and standard deviation among pixels.*

- P12, L4: For Figure S6, It would be better to show the 10 networks with different colors or symbols on the map... Also, this map should have been mentioned in section 2.4.
  *Response/Action: We changed the figure as requested and referenced it in Section 2.4.*

- P12, Figure 8: It is really hard to tell from the figure if there is any better performance of the downscaled products than the 9km SMAP product.
  *Response/Action: This figure summarizes the performance of different products with respect to temporal dynamics metrics. Since the downscaled product is derived from the 9km product, we do not expect to see significant improvement in temporal dynamics of soil moisture estimates. Rather we check to verify that the increased spatial resolution in the downscaled product does not decrease temporal performance. We emphasized this in the revised manuscript.*

- P12, Figure 8: Any idea why there is poor performance over some network (e.g., SNOTEL) but good performance over some other networks (e.g. SOILSCAPE). Why does SMAP 36km have the best agreement with iRON?
  *Response/Action: While this assessment is beyond the scope of this manuscript, we would like to note that while iRON has the best agreement with SMAP data*

*with respect to ubRMSE it does not have the best correlation. SOILSCAPE has the best correlation, but it has an average ubRMSE metric. More detailed assessment of SMAP retrievals across different ground truth stations is provided in:*
*Chan, S. K. et al. (2016) 'Assessment of the SMAP Passive Soil Moisture Product', IEEE Transactions on Geoscience and Remote Sensing, 54(8), pp. 4994–5007. doi: 10.1109/TGRS.2016.2561938.*

- P13, L7-9: This conclusion might only be true among the different NN downscaling schemes.
*Response/Action: We agree with referee, and changed this statement as following in the revised manuscript:*
*"Our investigation shows that topographic index and variability of NDVI within the coarse scale pixel do not provide additional useful information on the spatial heterogeneity of soil moisture for the downscaling using our proposed NN technique and are thus omitted in the final product."*

- P13, L20: "... this lowers the quality of the downscaling algorithm in bare soil or sparsely vegetated." Do any of the results support this statement? According to Figure 4, there is very high correlation coefficient over those regions.
*Response/Action: Lower quality is with respect to both correlation and bias. Figure 4 shows that those regions have high correlation, but Figure 7 shows the same regions have a negative bias in soil moisture estimates. We revised the sentence as following as also explained in response to an earlier comment:*

[revised manuscript text omitted]